# Full Soft Capacitive Omnidirectional Tactile Sensor Based on Micro-Spines Electrode and Hemispheric Dielectric Structure

**DOI:** 10.3390/bios12070506

**Published:** 2022-07-10

**Authors:** Baochun Xu, Yu Wang, Haoao Cui, Haoran Niu, Yijian Liu, Zhongli Li, Da Chen

**Affiliations:** Laboratory for Intelligent Flexible Electronics, College of Electronic and Information Engineering, Shandong University of Science and Technology, Qingdao 266590, China; xbcno12022@163.com (B.X.); wangyunn1119@163.com (Y.W.); 15376705493@163.com (H.C.); nhaoran2019@163.com (H.N.); zhonglili@sdust.edu.cn (Z.L.)

**Keywords:** full soft, hemispheric hills, CNT-sprayed, omnidirectional tactile, machine learning

## Abstract

Flourishing in recent years, intelligent electronics is desirably pursued in many fields including bio-symbiotic, human physiology regulatory, robot operation, and human–computer interaction. To support this appealing vision, human-like tactile perception is urgently necessary for dexterous object manipulation. In particular, the real-time force perception with strength and orientation simultaneously is critical for intelligent electronic skin. However, it is still very challenging to achieve directional tactile sensing that has eminent properties, and at the same time, has the feasibility for scale expansion. Here, a fully soft capacitive omnidirectional tactile (ODT) sensor was developed based on the structure of MWCNTs coated stripe electrode and Ecoflex hemisphere array dielectric. The theoretical analysis of this structure was conducted for omnidirectional force detection by finite element simulation. Combined with the micro-spine and the hemispheric hills dielectric structure, this sensing structure could achieve omnidirectional detection with high sensitivity (0.306 ± 0.001 kPa^−1^ under 10 kPa) and a wide response range (2.55 Pa to 160 kPa). Moreover, to overcome the inherent disunity in flexible sensor units due to nano-materials and polymer, machine learning approaches were introduced as a prospective technical routing to recognize various loading angles and finally performed more than 99% recognition accuracy. The practical validity of the design was demonstrated by the detection of human motion, physiological activities, and gripping of a cup, which was evident to have great potential for tactile e-skin for digital medical and soft robotics.

## 1. Introduction

In recent years, flexible sensors have attracted a lot of interest because of their potential in various bio-machines integrated applications including biosymbiotic [1], physiological measurement [2,3,4], bionic robotics [5,6], human-machine interaction [7,8,9,10,11], and so on. Compared with silicon-based devices with high hardness, the sensors manufactured with soft materials are more suitable for biomimetic tactile perception due to their good stretchability and conformality [12,13]. Various flexible materials such as polyethylene (PE) [14], polydimethylsiloxane (PDMS) [15,16], and polyurethane (PU) [17,18] have been widely applied to tactile sensors. Generally, these soft and flexible tactile sensors are based on sensing mechanisms including capacitive [19,20], piezoelectric [12], and piezoresistive [21,22]. Among these mechanisms, extensive research on capacitive has been widely carried out due to its ultra-high sensitivity, considerable stability, and economical efficiency. Generally, a capacitive pressure sensor consists of a simple structure (an elastomer dielectric between conductive electrodes), which converts pressure excitation into capacitance change. In this way, various micro-structures can be easily applied to both the dielectric and electrode for performance optimization.

Notably, the perception information of e-skin is limited by the device structure, and most of the current flexible pressure sensors can only detect external forces in the form of downward pressure. However, humankind could perceive not only longitudinal components but also sliding shear components of the actual contact force [23]. Omnidirectional force detection is particularly relevant for systems that have many degrees of freedom and the internal torque-sensing cannot always be accomplished. How to combine directional detection with the advantages of capacitive sensors is urgently put on the agenda. For that, several approaches have been designed to detect the force with their direction, which mostly involves the special dielectric structure and substrate with electrodes array.

Lee Hyung et al. [24] reports a capacitive tactile sensor array using flexible material for normal and shear force detection. However, measurement of the sensor shows that the full-scale range of a detectable force is only about 10 mN. Viry et al. [25] introduces a flexible capacitive three-axial force sensor made of conductive fabric electrodes, which is highly sensitive within a wide normal force range (up to 400 kPa) and touch-like tangential force ranges. However, the fabrication of the fabric electrode itself is complicated and the conductive stability has not been solved. For that, Zhenan Bao et al. [26] introduces a flexible electronic skin that can detect the pressure in different directions and the device is fabricated by a hill array whose design mimics the interlocked dermis-epidermis interface in human skin. However, the theoretical model of force directional detection has no clear statement.

Worth noting that all the devices above require micro-fabrication processes and nano-materials which inevitably induce the disunity of the sensor units and increase the cost. This is an important reason why the multi array-elements flexible omnidirectional force sensors remain in the laboratory and research institute. Machine learning, which has been widely studied recently, provides an appropriate way to solve the problem of disunity of multiple sensor units [27]. In the context of these research advances, the state-of-the-art developments expose two main challenges: on the one hand, eminent performance as well as the detection ability of 3D forces are equally necessary; on the other hand, as they are considered economical and industrial, the feasibility for scale expansion and low-cost fabrication are also indispensable.

In this work, a simple-manufactured full soft capacitive omnidirectional tactile (ODT) sensor was reported based on the micro-spines electrode and hemispheric dielectric structure. Compared with other angular geometries design, the hemispherical structure are isotropic in the XoY plane, which can truly achieve omnidirectional perception. Both theoretical analysis and practical experiments were implemented to design and demonstrate the sensor. Firstly, the finite element simulation is adapted to theoretically analyze the geometric design. When the force direction changes, the hemispheres are deflected towards it, resulting in the capacitance differential value between the adjacent sensing units. Secondly, when combined, the micro-spine with the hemispheric hills dielectric structure, a high sensitivity (0.306 ± 0.001 kPa^−1^ under 10 kPa), and a wide response range (2.55 Pa to 160 kPa) are achieved with the function of omnidirectional pressure detection. Moreover, to overcome the inherent instability in flexible sensors, machine learning approaches are introduced as a prospective technical routing. Several machine learning methods successfully classify force-direction changes, which finally achieved more than 99% recognition accuracy in a five-degree gradient of three different pressure. The practical validity of the design is demonstrated by several applications, including the detection of human motion, physiological activities, and the process of clamping. The evidence points to the great potential of tactile skin in digital healthcare and soft robotics.

## 2. Design and Analysis

The proposed sensor array is constructed in a sandwich structure, in which the MWNTS/PDMS coated stripe films are used as the electrode layer, and the hemispherical hill elastomer made by Ecoflex is used as the dielectric layer. Figure 1a schematically depicts the sensor array (6 × 6 capacitors), where the Ecoflex hemispheric elastomer is sandwiched between the upper and lower electrode layers and is laminated with Ecoflex adhesive. The Ecoflex dielectric layer with a low modulus (~600 kPa) can be used for better force transduction. Because of the better combination effect with the sprayed CNT, and although the modulus is a little bit higher (~1 MPa), PDMS is chosen as the electrode substrate. The top view of the sensor array is shown in Figure 1b, the radius of each hemisphere is 3.5 mm, and the distance between the adjacent hemispheres is 1 mm. This size specification can be flexibly adjusted according to the actual requirements. The upper and lower electrode stripes are intersected to construct simplified electrode array. According to this method, each hemispherical hill aligns and corresponds to four capacitor units.

In order to demonstrate the basic performance of our pressure sensor, a plane-parallel model of a sensor unit was prepared for capacitance testing. When the dielectric layer of the capacitor is compressed, the distance between the upper and lower plates of the capacitor is reduced, according to the capacitor formula
(1)$$C=\frac{\varepsilon S}{4\pi kd}$$ where ε is a constant decided by the materials of dielectric, S is the opposite area of the capacitor plate, d is the distance between the upper and lower plates, and k is the electrostatic constant of the plane-parallel capacitor. 

According to the capacitor model, the correctness of the omnidirectional theory was validated by finite element simulation. Figure 1c, d shows the stress distribution and z-axis deformation results under the action of normal force and inclined force, respectively. More detailed information of the calculation model and used parameters were given in Appendix A. There are cross-section diagrams with stress distribution and perspective view of the z-axis deformation under three different loading states, as the loading state of applying normal force (90°), 60°, and 30° force. When the normal force is applied, the four capacitors are subjected to the same stress, and the distance between the upper and lower electrodes is shortened to the same extent. In addition, the hemisphere is compressed vertically (Figure 1d) so that the capacitance could show the same changing trend. With the applied force tilting gradually, the hemisphere is squeezed as well, but the hemisphere leans to the same side, which leads to a capacitance-differential between the adjacent units. The larger tilting angle the force exerts, the larger the differential value obtained. So different force directions can be detected with aid of the capacitance trend presented.

## 3. Method

In order to simplify the operation and cost reduction, the conventional electrode preparation methods of photolithography and sputtering are abandoned. Based on a simple and cost-effective feature, the spraying process is adapted in this paper, during which a spray mask acted as the pattern template of the carbon nanotube coated electrode. 

Figure 2a,b shows the manufacturing process of each sensor component in detail. Firstly, the multi-walled carbon nanotube (MWCNT) suspension was prepared. Deionized water, MWCNT powder, and dispersant were mixed uniformly in proportion (49:1:1.6) and the mixture was dispersed by ultrasound for 3 h. The striped electrode template was adhesive on the ground glass wafer with a micro-spine structure. Then, the CNT suspension was uniformly sprayed and dried at room temperature. A PDMS mixture (PDMS:curing agent = 10:1) is prepared and then poured onto the silicon wafer after vacuum, so as to avoid the influence of bubbles when pouring. PDMS and CNTs were combined after half an hour of 80 °C vacuum curing of the wafer after spinning coating. The molecular attraction between CNT and PDMS is larger than the glass wafers. As the consequence, CNTs are easier to peel from glass wafers and adsorb on PDMS. The production process of the hemispheric hill medium layer consisted of several main steps as depicted in Figure 2b. Poured PVA solution into a 3D template, then spin-coated Ecoflex (A:B = 1:1). After natural drying and dissolving PVA in water, the hemisphere dielectric layer was fabricated. Assembled above components by Ecoflex adhesion, and a sensor array with variable scale was made.

The scanning electron microscopy (SEM) images was shown in Figure 2c,d with different multiples, respectively. The coated CNTs layer can be observed from the section view in Figure 2c. Figure 2d depicts the uncoated portions where the bare PDMS is. After spraying, the rugged structure distributes irregularly on the surface from the micro-view, but the distribution has good consistency on the entire scale of surface from the macro-view. Micropores appear in the composite material caused by spraying at the surface of micro-caves, and MWCNTs extend out of the nanocomposite matrix in a disorderly way and circled out by the red dots inset. This reveals the anfractuous and intertwining arrangement of MWCNTs in the nanocomposite results in a large number of intricate conductive networks around the nanocomposite and generates more conductive channels.

Figure 2e gives the photo of the fabricated sensor. The overall optical image of the tensile measuring instrument, multimeter, and LCR measuring instrument used in the tests is shown in Figure 2f, and detailed information and parameters were given in Appendix A. The LCR meter with the detection accuracy of 10^−4^ pF, which is sufficient accuracy for the tested capacitance of the sensor (~1 pF).

## 4. Results

### 4.1. Fundamental Performance Test

Usually, this soft sensor operates in the state of stretching and bending in many cases. On account of the MWCNTs coated layer being on the surface of PDMS, it is necessary to check whether the electrode is stable under the above conditions. The resistance of four paralleled strip electrodes have been measured under the increasing strain from 0 to 50%. As shown in Figure 3a, the resistance of the electrodes increase steadily with the strain, which may be caused by the increasingly sparse conductive path.

Under the condition that the plate area stays stable and the dielectric constant remains unchanged, three weights (400 Pa, 800 Pa, 2 kPa) were repeatedly applied on the sensor to demonstrate the practical repeatability as Figure 3b shows. The experiment showed that twice loading and unloading results in a stable capacitance change without hysteresis. Appendix A shows the fast response of the ODT sensor, as well as the response time (0.15 s) and recovery time (0.1 s). These values are faster than the general response time of the human brain through conscious processing (~0.2 s) [28]. Furthermore, since the measurement of this parameter depends on the time accuracy of the LCR meter (can maximum record 20 points per second), it means the temporal resolution of the device is 50 ms. In addition, the process of placing and lifting weights inevitably takes some time, so that the actual response time of our sensor is even less than the given values (0.1~0.15 s). Its stable response and recovery characteristics ensure the application prospects in bionic robotics and human-machine interfaces. 

Both subtle and high pressure sensitivity should be represented, and the tensile machine is used to continuously apply pressure to the sensor up to 160 kPa, much higher than the typical human touch sensing range (~10 kPa). For a parallel-plate capacitor, the sensitivity is directly proportional to the compressed distance (Δd) of the dielectric layer and can be further expressed as [29]:
$$S=\frac{\Delta C/C_{0}}{\Delta P}=\propto \frac{\Delta d/d}{\Delta P}$$ where *d*_0_ is the distance of the parallel electrodes. This distance is dependent on the thickness of the dielectric layer, which indicates an inverse relationship between the equivalent modulus *E* and pressure sensitivity *S*. With increasing pressure, the hemispheric dielectric structure collapses, which leads to the continuous increase of the equivalent modulus *E* [6]. Consequently, the sensitivity curve shows an deceleration of increase. The measured sensitivity shown in Figure 3c was 0.306 ± 0.001 kPa^−1^ under low pressure (0–10 kPa) and 0.020 ± 0.002 kPa^−1^ in the range *p* > 60 kPa. Moreover, the ground glass could give micro-spinous microstructures to the electrode [29], so that the minimum detection limit was no more than 10 mg (2.55 Pa) in our test (Figure 3d), which makes the device more accurate in practical application.

Figure 3e is the fatigue testing of 3200 cycles of compression release applied to the sensor in the tensile machine to evaluate the sensor reliability. The sensor has good long-term stability and mechanical durability. To attest, the sensor is more widely used in different circumstances, and the sensor was tested under different temperatures. It is important to select the same capacitor to ensure that the effect is unified, and the environment temperature was increased from room temperature (23 °C) gradually to 50 °C. Worth noting that the capacitance fluctuation was no more than 1%, even when the temperature increases are out of the survival condition (Figure 3f).

### 4.2. Experiment of Omnidirectional Force

In order to characterize the force directions, the adjacent pairs of capacitors right above the hemispherical hill were the focus of the research. The same force intensity was applied to the sensor using the pressure gauge for the normal force (90°) applied, and the 60° and the 30° inclined tables were used to bring the directions. Figure 4a shows a top view of the capacitor array, and the position of coated CNTs electrodes are marked with their numbers. Figure 4b shows the difference of capacitance variation of adjacent units corresponding to 2 KPa pressure inclined along the XoZ plane at different angles. For the sake of observation, the absolute value (|C_Bx_ − C_Ax_|) of the part less than 0° (shadow area) is taken to obtain a positive and negative angular symmetric curve. The approximate symmetry of the curves in the range of −60° ≤ α ≤ 60° reflects the consistency between the counter directions. Additionally, each pair of capacitors shows a similar increase trend.

In order to further simplify the expression of data to facilitate analysis, the relative variation $R_{A1}$, $R_{A2},\;R_{B1},\;R_{B2}$ of the capacitor is defined and calculated as follows.
(2)$$R_{A1}=\frac{\Delta C_{A1}}{\Delta C_{A1}+\Delta C_{A2}+\Delta C_{B1}+\Delta C_{B2}}$$
(3)$$R_{A2}=\frac{\Delta C_{A2}}{\Delta C_{A1}+\Delta C_{A2}+\Delta C_{B1}+\Delta C_{B2}}$$
(4)$$R_{B1}=\frac{\Delta C_{B1}}{\Delta C_{A1}+\Delta C_{A2}+\Delta C_{B1}+\Delta C_{B2}}$$
(5)$$R_{B2}=\frac{\Delta C_{B2}}{\Delta C_{A1}+\Delta C_{A2}+\Delta C_{B1}+\Delta C_{B2}}$$

$\Delta C_{A1},\;\Delta C_{A2},\;\Delta C_{B1},\;\Delta C_{B2}$ denote the change in a single capacitor value, respectively. The corner marks correspond to each capacitor as shown in Figure 4, which is crossed by upper and lower strip electrodes. Figure 4c–e shows the capacitance hot spot maps under different pressures and angles which is represented by the relative variation (R). As the normal pressure (90°) increases gradually, the hemispheric hill structure is squeezed more significantly, but the relative variation R corresponding to four capacitors remained unchanged. Figure 4d shows the 60° inclined force of 400 Pa, 800 Pa, and 2000 Pa. Notably, with the increasing pressure applied, the four capacitors are compressed to different depths. The compression depth of the capacitor on the inclined side is higher (the relative variation R ≈ 0.3) than that on the other side (R ≈ 0.2). The gravity center of the hemisphere shifts to the inclined direction, which conforms to the mechanical principle and theoretical simulation. Figure 4e shows the applied force of 400 Pa, 800 Pa, and 2000 Pa inclined to 30 degrees. The calculation method and analysis method are the same as the inclined force with 60 degrees. As the pressure increases gradually, the center of gravity moves down to the inclined side more obviously (R ≈ 0.35), to distinguish different directions. On some maps, however, we can observe an asymmetry along the Y-axis. This is due to the inherent instability of flexible device fabrication, and the initial capacitance and pressure change rate of the four capacitor units are not uniform. This inherent inconsistency is inevitable because of the use of nano-materials and polymers. Furthermore, this disunity increases the difficulty of feature extraction when directly analyzing data, which is an important reason for us to introduce machine learning.

The process of sorting out the relative change rate of a large number of data points seriatim is complicated and time-consuming. Additionally, the inevitable instability and disunity of the multi-units flexible sensors make it difficult to directly analyze a bulk of data paths. Thus, as an potential technical routing, the burgeoning machine learning was introduced, aiming to achieve efficient prediction by inputting bare multichannel capacitance data. Figure 5a–c respectively shows the scatter diagram of data distribution under 16 conditions of four pressure loads (0, 400, 800 and 2000 Pa) at five angles (−60° ≤ α ≤ 60°) of three pairs of electrodes mentioned above. As can be seen from the scatter diagram, when there are only two channels input, the data viscosity is relatively large. This leads to traditional classification methods and a high misjudgment rate. However, when the data paths reach six channels, the accuracy of force direction identification is close to 100% under the cross verification of three pairs of features. 

In the experiment, we used several common machine learning methods for verification and selected three of them with better feature recognition effects show in Figure 5d–f. Among them, the Gaussian naive Bayes classifier and Gaussian kernel support vector machine (SVM) only have 1 set of data error predictions in 600 sets of data (99.83% accuracy). The artificial neural network had a higher inference speed but showed 12 groups of prediction errors (98% accuracy). Overall, our ODT sensor can be effectively combined with nowadays rapidly developing data processing methods. Although there is still a need for the further development of the algorithm mechanism, this experiment provides a favorable technical foundation for the next generation of intelligent tactile sensing.

### 4.3. Application Test

Due to their excellent performance, our ODT sensors have the ability to detect a variety of forces, including but not limited to pressure and bending forces. 

As Figure 6a shows, when the device is bent at different degrees, the single capacitor shows a relatively stable and obvious difference. The dielectric layer is compressed with increasing bend angles, and the distance between the electrodes of the capacitor decreases, resulting in the capacitance value increases. These results indicate that our pressure transducer in biomedical monitoring of the human body has many potential application prospects, has convenient flexibility, and is non-invasive. Furthermore, applications in humans are designed with the aim of proving the sensor’s shape conformality, stretch and bend reliability, as well as its feasibility for soft bionts. The fabricated sensor was wholly basted on human skin by the PU tape. With the benefits of a relatively low modulus (104~106 Pa) and thickness (5 μm) [29], the use of PU tape hardly forces the sensor. Pressure sensors were installed on the outside of the index finger as depicted in Figure 6b, and used to detect the joint flexion. Once the pressure sensor is squeezed, the capacitance value increases. When placed at the wrist for monitoring (Figure 6c), the wrist was initially kept level (Appendix A), and the upward bending resulted in a reduction, whereas the downward bending resulted in increased capacitance. It was also installed on the outside of the elbow and the outside of the knee for more all-around monitoring (as shown in Figure 6d–e), which further certified the wide application of our sensor after a simple and low-cost manufacturing process.

To monitor the movement of picking up the cup, the two capacitors (A_1,_ B_1_) of the sensor pasted on the outside of the glass cup were contrasted, and the gesture was shown in Appendix A. Through repeating the same height, the friction and grip force form a downward sloping force. The lower sensor (B1) changes more significantly than the higher sensor (A1), that is, the pressure of the lower electrode is higher than that of the upper electrode. This test demonstrates the potential applications of sensors for future precision operations. 

## 5. Conclusions

From what has been discussed above, a simple-manufactured full soft capacitive omnidirectional tactile (ODT) sensor was demonstrated based on the MWCNT-coated micro-spines electrode and hemispheric dielectric structure. According to the finite element simulation, theoretical analysis of the geometric design was testified. When combining the micro-spine with the hemispheric hills dielectric structure, high sensitivity (0.306 ± 0.001 kPa^−1^ under 10 kPa) and a wide response range (2.55 Pa to 160 kPa) were achieved with the function of omnidirectional pressure detection. Moreover, machine learning methods were introduced to classify and process multiple sets of measurement data, and more than 99% recognition accuracy was finally achieved. The validity of the design was demonstrated by the detection of human motion, physiological activities, and the process of clamping, which was evident to have great potential for tactile e-skin for digital medical and soft robotics.

## Figures and Tables

**Figure 1 biosensors-12-00506-f001:**
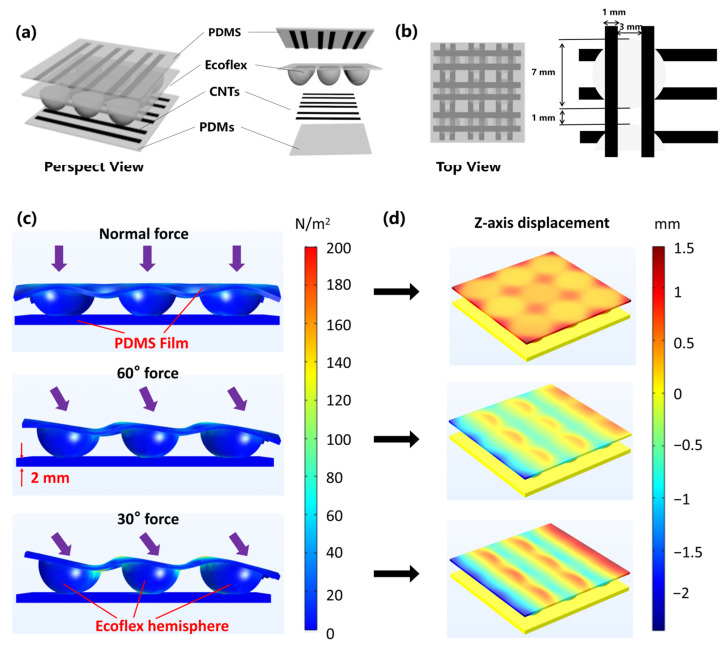
**Flexible omnidirectional tactile sensors consisted of MWNTS/PDMS coated films and hemisphere array.** (**a**) Schematic diagram of the sensor structure in perspective view, including upper and lower electrodes and the dielectric layer. (**b**) Top view of the sensor with the size marking. (**c**) Simulation schematic of the ODT sensor under different directions of 1N forces with the informative marking. (**d**) Simulation of the z-axis variable of electrode layer corresponding to different direction pressure.

**Figure 2 biosensors-12-00506-f002:**
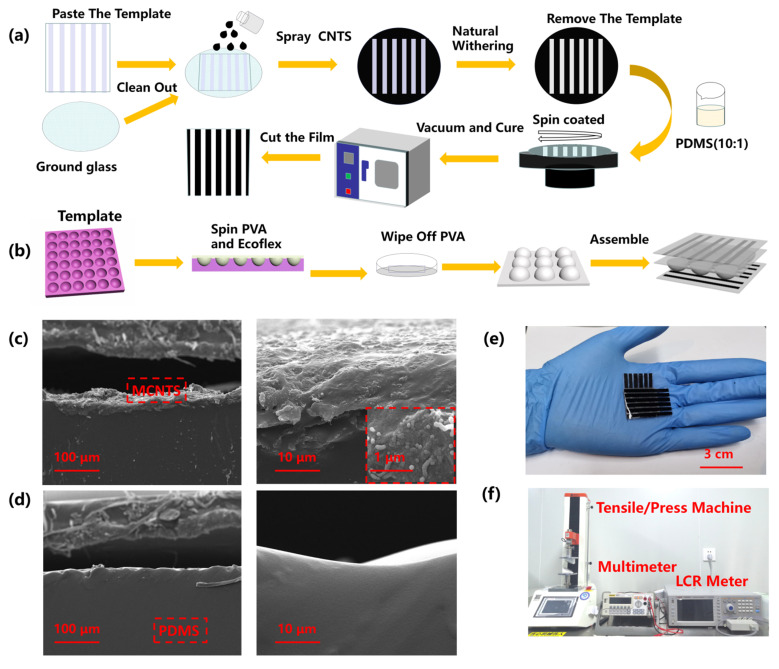
**Fabrication process and experiment platform of the ODT sensor.** (**a**) The fabrication process of the micro-spined stripe electrodes of the pressure sensor. (**b**) The fabrication process of the hemispherical hilly dielectric layer. (**c**) SEM images from the section view of nanocomposite materials under different magnification, focusing on the MWCNT conducting layer. (**d**) SEM images from the section view under different magnifications, focusing on the bare PDMS microstructure. (**e**) Optical photograph of the soft ODT sensor. (**f**) Optical photograph of the testing system during the experiment.

**Figure 3 biosensors-12-00506-f003:**
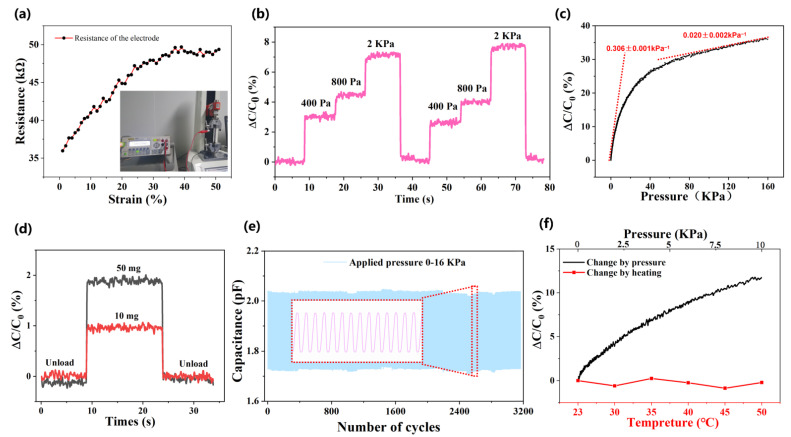
**Fundamental performance test of the soft omnidirectional tactile sensor.** (**a**) The resistance stability of the coated electrodes during a 50% tension, with the measuring data point (black point) and fitting curve (red curve). (**b**) The dynamic capacitance response of the sensor under repeated various pressures (400, 800, 2000 Pa). (**c**) The capacitance response curve of the flexible pressure sensor under continuous pressure (0~160 kPa). (**d**) The optimized devices response of micro pressures of 50 mg and 10 mg. (**e**) Long-term durability test (over 3200 cycles) of the flexible pressure sensors at a pressure of 16 kPa. Inset: magnified view of 15 cycles for the late stage of the cycle (around 2400 cycles). (**f**) The stability curve (red broken line) of the flexible pressure sensor with the increasing temperature and the capacitance of the sensor increases gradually with the increase of pressure (black curve).

**Figure 4 biosensors-12-00506-f004:**
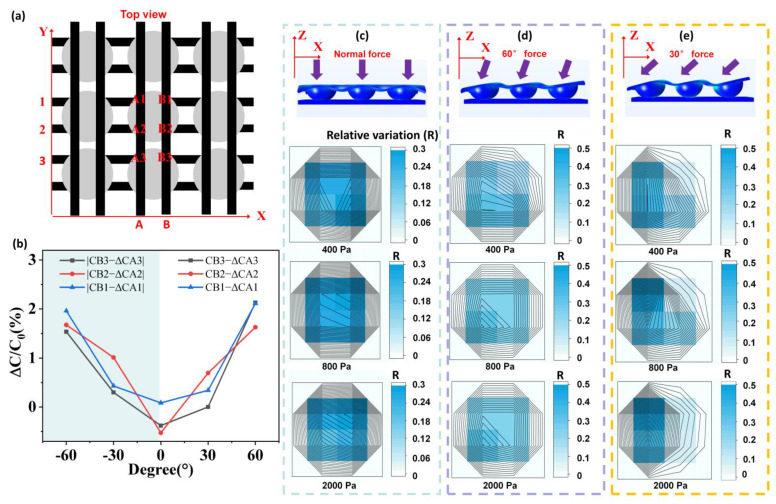
**Representation of forces in different directions by ODT sensors.** (**a**) Schematic diagram of electrode distribution set to be measured and characterized. (**b**) The difference of capacitance variation of adjacent units corresponding to 2 KPa pressure inclined along XoZ plane at different angles. (**c**–**e**) Top: Cross-sectional views with forces shown in arrows (normally, 60° and 30° incline, respectively). Below: Top views of the relative changes in capacitance shown for capacitors applied by different directional forces with the pressure at 400 Pa/800 Pa/2000 Pa.

**Figure 5 biosensors-12-00506-f005:**
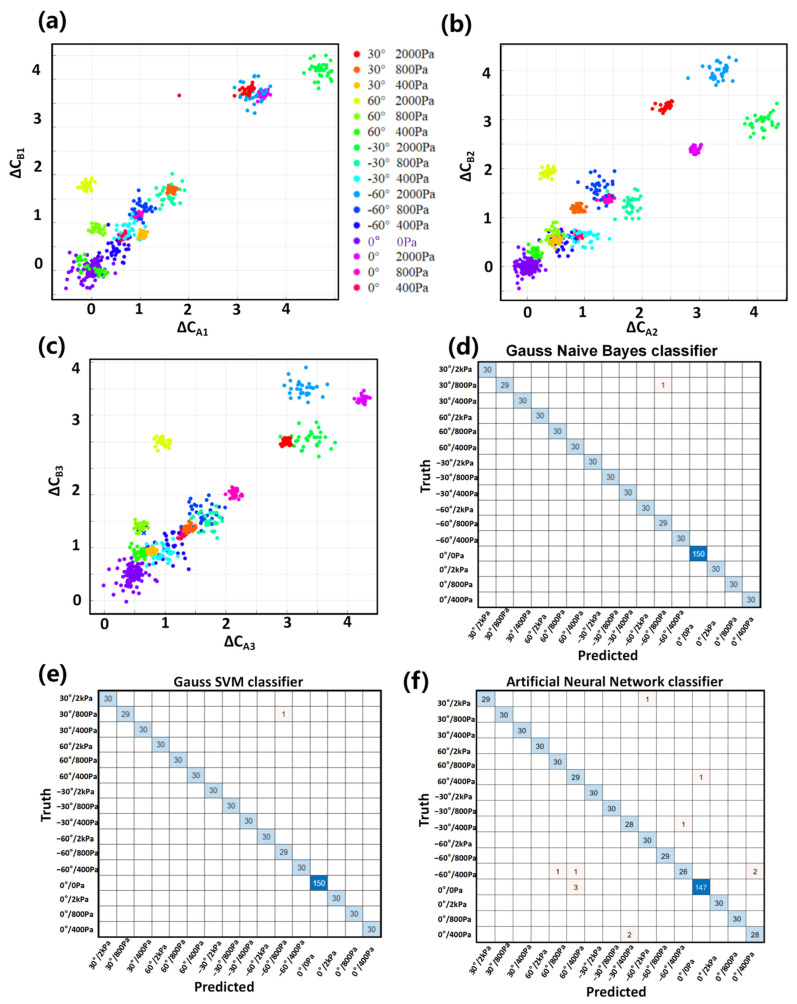
**Results of classification methods based on machine learning.** (**a**–**c**) Six channels of data from above mentioned six electrodes in Figure 4a, which were characterized in pair with their relatively variable. (**d**) The machine learning recognition result is based on the Naive Bayes classifier, which has a high accuracy of 99.83%. (**e**) The machine learning recognition result is based on the Gauss SVM classifier, which also has a 99.83% accuracy. (**f**) The machine learning recognition results are based on the Artificial Neural Network classifier, which has slightly lower accuracy (98%).

**Figure 6 biosensors-12-00506-f006:**
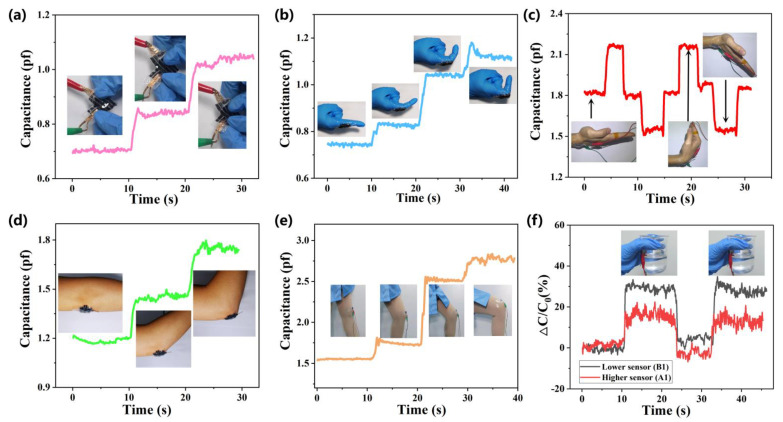
**Experimental results of various mechanical stimulate stresses which were extracted from human body testing.** (**a**) Capacitance change curves of flexible pressure sensors with different tortuosity (without torture, around 45°torture, and 90°, respectively). (**b**–**e**) The capacitance curves of the joints as the finger, wrist, elbow, and knee with the change of bending degree are pasted on the device. As the flexion of the joints varies, the signal shows different strengths. (**f**) Paste on the outside of the cup to detect the signal changed curves. The pink line stands for the lower electrode which changed more acutely and the purple line is the upper electrode with less variation.

## Data Availability

The data that support the findings of this study are available from the corresponding author upon reasonable request.

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
