# Peer review of "Full Soft Capacitive Omnidirectional Tactile Sensor Based on Micro-Spines Electrode and Hemispheric Dielectric Structure"

_biosensors, 2022, doi:10.3390/bios12070506_

Round 1

Reviewer 1 Report

Major issues:

1. The general phrasing should be improved as it seems not accurate enough in several instances (words repetition in the same sentence, telegraphic style randomly occurring in the methods, etc.). The authors should also double check for the many typos present (e.g. double spaces or lack of space after special characters as “=”). Description of results should be improved, as it is now very cryptic, also discussing some non predicted behaviors that emerge from the data (see below).

2. It is not clear the aim of the classification analysis, since it is not based on recognizing specific textile or stimulation patterns, but only pressure and inclination conditions, which are theoretically continuous (the values are a priori selected by the authors). Instead, a more logic approach would be to calibrate the instrument and provide some calibration curve for inclination and pressure (as it was partially done in the previous analyses). Classification approaches should be adopted for the later experiments (e.g. stretching movements of the human leg and hand), although the data seem to be poorly informative about the specific pattern of deformation.

3. The experiments described with human movements are difficult to be reproduced and only a few cases are provided. Also, it is not clear how the positioning and fixation of the device was performed. In some cases (e.g. sensors are not visible in the beaker photo). These images and descriptions should be strongly improved.

Other comments to the paper:

The authors should better explain why this device would be better compared to other that are produced using different techniques, including more traditional sensors.

line 123: figure title should not coincide with a panel title.

line 173: what do authors mean by “the conductivity of the electrode does not break”? Better specify the expected behavior. Actually the graph shows a very noisy behavior of the device over about 18% strain rate (please also define this measure) that must be discussed, and the fitted plateau is nor really reliable apparently.

Line 177: “constantly increased pressure” what do the authors mean by constant and why have those pressure values been selected (no linearly spaced)?

Line 180: Figure S1 was not provided (no supplementary information was uploaded)

Line 182: The sentence about human brain conscious processing has to be revised, since it appears as too naive. Please also provide reference to these considerations.

Line 183: the sentence seems specious: reading of capacitance will always introduce some limitations and authors should provide exact specifications of the reading device to infer better performance potential for the device.

Line 185: Which effect?

Line 238: “as the tilt of 30 degrees” 30 is likely a mistake, do the authors mean “60”?

Line 241: Rephrase the sentence (which is the subject of “impose”?)

Line 242: What does “deviation” refers to? What pressure value was used for Figure 4b? All this paragraph has to be rephrased.

Figure 4: panels letters should be placed above (not below) the panels. Figure style is also a bit confusing (dashed lines around panels).

Figure 4b: at which pressure value are these data obtained?

Figure 4e (400 and 800 Pa): please discuss the asymmetry in the map on the “Y” axis which is observed only in these conditions

Line 252: The pictures of the electrodes are blue colored (not black)

Author Response

Dear reviewer,

Thanks for your careful review. According to the reviewer's comments, we carefully revised the manuscript and replied the comments one by one. Please see the attachment.

Best regards,

Baochun XU

Reviewer 2 Report

This manuscript by B. Xu et al. presents a soft capacitive omnidirectional tactile sensor by using a structure made of MWCNTs coated stripe electrodes and Ecoflex hemisphere array. Direction-resolved pressure detection with sensitivity of 0.306kPa-1 and a wide response range between 2.5Pa and 160kPa were demonstrated. The sensors were also successfully applied in several cases for detection of human body movements. The obtained results are promising in the development of e-skin devices and technologies, and I therefore recommend its publication in Biosensors. I have several comments as given below.

1. In the introduction part, the research gap is not clear enough. There are many flexible capacitive force sensors reported in literature. It is not clear about the contribution or uniqueness of this work. The authors need to clearly discuss more about the novelty behind the work.

2. As related to my above comment, the sensor design concept is unclear. Why are the materials of MWNTS/PDMS and Ecoflex used? Also why is the dielectric layer designed to be hemisphere in shape? Any consideration on the geometric sizes? An elaboration on the design approach would helpful.

3. Fig. 1 presents the finite element simulation results. An detailed information on the calculation model and used parameters need to be given for a complete understanding on the results.

4. In Fig. 3, why is the sensitivity at low pressure higher than that at high pressure. A analysis on the dependence of sensitivity on pressure range is needed.

5. Some language errors need to be corrected. For examples: (1) Page 6 lines 214-216: “In order to characterize the sensor's resolution of tactile directions, the four capacitors corresponding to the hemispherical medium in the middle of the sensor as the main parameters. ”; (2) the label in Fig. 4 “Relative variatio (R)” should be “variation”.

Author Response

Dear reviewer:

Thanks for your careful review and positive comments for this paper. According to the reviewer's comments, we have carefully revised the manuscript and replied the comments one by one. Please see the attachment.

Best regards,

Baochun XU

Round 2

Reviewer 1 Report

The authors extensively addressed the issues raised.